# Does the Metabolome of Wild-like *Dendrobium officinale* of Different Origins Have Regional Differences?

**DOI:** 10.3390/molecules27207024

**Published:** 2022-10-18

**Authors:** Qiqian Lan, Chenxing Liu, Zhanghua Wu, Chen Ni, Jinyan Li, Chunlei Huang, Huan Wang, Gang Wei

**Affiliations:** 1School of Pharmaceutical Science, Guangzhou University of Chinese Medicine, Guangzhou 510006, China; 2Shaoguan Institute of Danxia Dendrobium Officinale, Shaoguan 512005, China; 3Hantai Biomedical Group Co., Ltd. Zibo Br, Zibo 255000, China; 4Hunan Institute for Drug Control, Changsha 410001, China

**Keywords:** *Dendrobium officinale*, widely targeted metabolomics, different origins, flavonoids

## Abstract

*Dendrobium officinale*, as a traditional Chinese medicine, has considerable commercial value and pharmacological activity. Environmental factors of different origins have a great influence on *Dendrobium officinale* metabolites, which affect its pharmacological activity. This study sought to identify the differential metabolites of wild-imitating cultivated *D. officinale* stems of different origins. Using the widely-targeted metabolomics approach, 442 metabolites were detected and characterized, including flavonoids, lipids, amino acids and derivatives, and alkaloids. We found that although the chemical constitution of *D. officinale* cultured in the three habitats was parallel, the contents were significantly different. Meanwhile, the KEGG pathway enrichment analysis revealed that the distinctive metabolites among the three groups were mainly involved in flavone and flavonol biosynthesis. To further explore the different contents of flavonoids, HPLC was performed on four main flavonoid contents, which can be used as one of the references to distinguish *D. officinale* from different growing origins. In conclusion, a comprehensive profile of the metabolic differences of *D. officinale* grown in different origins was provided, which contributed a scientific basis for further research on the quality evaluation of *D. officinale*.

## 1. Introduction

*Dendrobium officinale* is a plant of the orchid family with a distribution range of nearly 1400 species and a rich medicinal value [1]. *D. officinale* is known as the “Chinese Immortal Herb” in China. It was first recorded in *Shen Nong’s Materia Medica*, which described *D. officinale* growing on rocks, and has been used in medicine for 2000 years. As a treasured Chinese medicinal ingredient, *D. officinale* is rich in polysaccharides, flavonoids, alkaloids and other bioactive substances, which play important pharmacological roles in anti-diabetic, anti-cancer, antioxidant and anti-fatigue [2,3]. *D. officinale* has spread to a considerable number of countries around the world, such as Australia, Japan, and the United States, and it is more commonly distributed in China [4]. However, due to the relatively slow growth rate, natural climate change, habitat destruction, and overexploitation, wild *D. officinale* resources are rapidly declining and are facing the threat of extinction [5]. Therefore, it is crucial to advance the artificial cultivation of medicinal *D. officinale*. In recent years, to meet the market demand and protect wild resources, researchers have tried to grow *D. officinale* using artificial cultivation methods [6]. However, another cultivation method, which is different from artificial cultivation, imitating wild cultivation on stones, is becoming another potential cultivation method [7]. Compared with the previous artificial cultivation, imitating wild cultivation on stones is closer to the wild living environment of *D. officinale*. Hence, in this study, wild *D. officinale* imitated on stones was used as the material for metabolomic analysis. It is widely accepted that components such as polysaccharides, flavonoids, and alkaloids are the main sources of benefits generated by *D. officinale* [8], and all of these components fluctuate with changes in the growth environment (humidity, light duration, and temperature) of *D. officinale*. Therefore, it is important to explore the differences in the composition of *D. officinale* from different origins and identify the optimal growth environment to improve yield and quality.

Metabolomics has been extensively developed in herbal research identifying the authenticity of medicinal plants [9], determining the harvest time of raw medicinal materials [10], exploring the effects of soil characteristics on changes in the plant metabolome [11], screening processing methods [12] and other factors. Metabolomic approaches provide valuable resources for the identification of metabolite differences between mutant and wild-type plants, the identification of metabolite changes in original herbs, and the quality control of herbal medicines [13,14,15,16,17]. Metabolomics is based on high-throughput tools that allow the simultaneous qualitative and quantitative analysis of many metabolites and the discovery of correspondences between metabolites and physiological changes [18]. Therefore, it can be used to detect changes in metabolites in plants under different growth environments. Briefly, metabolomics is a more powerful technique because metabolites and their concentrations directly reflect the potential biochemical activity and metabolic state of cells, tissues, or organisms [19,20,21]. Numerous studies have reported extensive metabolic information on the stems and leaves of *D. officinale*. Using metabolomic analysis of greenhouse *Dendrobium* stems of plants grown for a different number of years, Yuan et al. found that the accumulation of flavonoids was higher in two-year-old *D. officinale* stems [22]. Using greenhouse six-month-old *Dendrobium* stem and leaf metabolomic data, Cao et al. reported that both organs contained similar metabolites, but the metabolic concentrations were different [23]. Zuo et al. analyzed the metabolome of greenhouse *Dendrobium*, and they discovered that the chemical composition of *Dendrobium* cultured in three substrates was similar, but the contents were significantly different [24]. The abovementioned studies were only analyzed by metabolomics of different growth years, organs, and cultivation substrates of *D. officinale*. Nevertheless, there is no report on the comprehensive analysis of the chemical and metabolic composition of imitating wild-cultivated *D. officinale*. Since ancient times, the medicinal *D. officinale* has been wild and mostly presents rust-colored spots, while it is observed that the appearance of *D. officinale* in greenhouses is mostly green. Therefore, we collected three types of wild-like *D. officinale* samples from Guangdong, Zhejiang, and Guangxi Provinces and performed metabolomic analysis on them.

*Dendrobium* has been shown to possess medicinal properties primarily in the stems [25]. Therefore, wild imitating cultivated *D. officinale* stems of different origins were selected as experimental materials, and the metabolomics and HPLC analyses were combined to compare them. By comparing their chemical compositions from the three main origins, the differences in their compositions were revealed, which was vital to comprehensively evaluate *D. officinale*. In this work, the UPLC–MS/MS system combined with multivariate numerical analysis was expanded to construct the metabolic profile of wild-like *D. officinale*. In the process of simulating wild-cultivated *D. officinale*, the variations in the arrangement and substance of metabolites, including flavonoids, alcohols, organic acids, amino acids, and other metabolites, were studied to identify biomarkers. At the same time, using the UPLC approach, a stable and simple method can be used to quickly distinguish *D. officinale* from different origins, which provides an orientation for the identification and control of the quality of wild-like *D. officinale*.

## 2. Results

### 2.1. Habitat Survey

In order to further investigate the habitat of wild *D. officinale*, we conducted an inspection of the habitat of wild dendrobium in Danxia landforms of Guangdong, Guangxi, which is one of the traditional genuine producing areas. The wild *D. officinale* plants on the Danxia stone walls were rust-colored (Figure 1). Compared with the three origins of wild-like *D. officinale* (Figure 1), the appearance of the RS samples was more similar to wild *D. officinale*.

### 2.2. Data Evaluation

To assess the stability of the instruments and repeatability of the samples, Pearson’s correlation coefficients were computed with the “cor” function in R (Ver. 3.5.0). The results of the correlation analysis are given in Table 1. The comparisons of GP vs. RS, RS vs. ZJ, and ZJ vs. GP exhibited a high correlation coefficient (0.83320, 0.92307 and 0.82796, respectively), showing that the metabolites screened in the three pairs of comparative groups were reliable. Correlation coefficients between intragroup samples (GP vs. GP, RS vs. RS, ZJ vs. ZJ) were higher than those in intergroup samples (GP vs. RS, RS vs. ZJ, ZJ vs. GP), which illustrated that the stability of the instruments and the reproductivity of samples were good. Therefore, the method was suitable for qualitative and quantitative analyses.

### 2.3. Principal Component Analysis

We performed principal component analysis (PCA) on the data to obtain a prior understanding of the overall metabolome. The metabolic distribution trend among different groups and the degree of discreteness of different molecules in each group are shown in Figure 2. The scatter points corresponding to the four samples in the RS, GP, ZJ, and MIX groups were clustered together within the group, indicating that the repeatability within the group was relatively good, and there was a good degree of discrimination among the four groups (Figure 2a). PC1 and PC2 explained 31.3% and 20.79% of the metabolite variations among all samples, respectively, representing 52.09% of the total metabolic variation. Quality control (MIX)samples were gathered in a group and separated from others, which showed that QC samples had good repeatability. As shown in Figure 2b, the ZJ sample was significantly separated in the direction of PC3. The overall results indicated that the three groups of samples were well-distinguished.

### 2.4. Orthogonal Partial Least Squares Discriminant Analysis

Although PCA can effectively extract the main information, it is not sensitive enough to exclude the variables with smaller correlations. The use of OPLS-DA can solve this problem, and the differences between groups can be expressed to the greatest extent. To validate the OPLS-DA models, 200 permutation trials were performed in this study. Q2 and R2Y are vital limitations for assessing the example in OPLS-DA. In this report, the Q2 and R2Y values of the GP vs. ZJ, GP vs. RS, and RS vs. ZJ models were 0.998 and 0.922, 1 and 0.961, and 0.999 and 0.903, respectively (Figure 3). The Q2 and R2Y values of all contrast groups exceeded0.9, indicating that these models are dependable and stable. Meanwhile, it can be said that these models can be used to further screen different metabolites. The OPLS-DA score plots are shown in Figure 4. Samples from the GP vs. ZJ, GP vs. RS, and RS vs. ZJ groups had a clear degree of separation, which illustrated that the metabolites generally exhibited significant differences among the different production areas.

### 2.5. Widely Targeted Metabolomics Profiling

To study the variations in metabolites from the different origins of *D. officinale*, we performed widely targeted metabolic profiling on samples of *D. officinale* stems. According to UPLC–ESI–MS/MS, *D. officinale* metabolites in the three main production areas were studied. A total of 442 metabolites were identified, and they were divided into 16 major classes, including flavonoids, lipids, amino acids and their derivatives, alkaloids, phenolic acids, nucleotides and their derivatives, organic acids and lignans and coumarins, and others (Table 2). Among them, the top three most represented metabolites were flavonoids, lipids, amino acids and their derivatives, which contained 102, 72, and 53 metabolites, respectively. However, there were almost no compounds belonging to quinones, proanthocyanidins, tannins and steroids in *D. officinale* stems. Generally, flavonoids are the main components of *D. officinale* metabolites, with eight types of flavonoids, including flavanones, flavones, flavonols, isoflavones, flavonoid carbons, chalcones, and anthocyanins. It accounted for 1/4 of the total detected metabolites (Appendix A). The comprehensive metabolic profile in the mature stalks of wild-imitating cultivated *D. officinale* obtained in this report will help the comparison of crucial biologically active ingredients among species. In addition, because most of the metabolites acknowledged in this research have not been reported in the wild imitating cultivated *D. officinale*, our research can provide forecasts for the discovery of new biologically active compounds.

### 2.6. Identification of Differential Metabolites

Metabolomics data are high-dimensional and massive; thus, a combination of univariate analysis and multivariate statistical analysis is needed to ultimately and accurately mine for differential metabolites. Detailed information on the differential metabolites among the groups is shown in Appendix A; the top 10 up and downregulated metabolites among these groups are shown in Appendix A. As shown in Appendix A, it is preliminarily concluded that Cis-4,7,10,13,16,19-docosahexaenoic acid detected in the ZJ group was the same significantly different metabolite compared to the RS and GP groups. For the RS group, 9,10- dihydroxy -12-octadecenoic acid was the same significantly differential metabolite as the other two groups. The differences between the three comparison groups were predominantly related to the synthesis of flavonoid components with apigenin and quercetin as the parent nuclei. The volcano plots can visually view the difference in the expression levels of metabolites in the two samples and the statistical significance of the difference (Figure 5).

Figure 5a shows that there were 166 metabolites with significantly different contents between the GP and RS groups. Comparing the two results (RS vs. GP), 48 metabolites were upregulated, and 118 metabolites were downregulated. A total of 137 metabolites were highly different in content between the GP and ZJ groups (Figure 5b), among which 84 metabolites increased, and 53 metabolites decreased in content in the ZJ group compared to the GP group. A total of 125 metabolites were recognized as significantly different between RS and ZJ (Figure 5c). Of these metabolites, there were 37 metabolites with significantly higher content in ZJ than RS, while there were 88 metabolites with lower content.

Venn diagram analysis, which is based on differential metabolites that were significantly different in the three comparative groups, shows the prevalent or unique metabolites present in the group. The abovementioned graph shows that a total of 244 metabolites that existed in at least two groups were shown to clearly change in content (Figure 6). Meanwhile, there were 22 significantly different metabolites shared among the three groups, and the comparison groups had unique differential metabolites (Table 3). The heatmap of the different metabolites commonly found in the three comparison groups (Figure 7b) also showed differences. Thus, the differential metabolites can be clearly distinguished between GP, RS, and ZJ samples. We found that the three groups of significantly different metabolites were mainly from flavonoids, lipids, and amino acids and derivatives (Figure 7a). The obtained results showed that compared to the GP and ZJ groups, the flavonoid content in the RS group significantly increased. Furthermore, lignans and coumarins were most abundant in the ZJ group. In addition, the expression of amino acids and their derivatives, organic acids and nucleotides and derivatives was higher in the RS group than in the other two groups. The distribution of all metabolites that varied significantly in content indicated that flavonoids were the most fluctuating metabolites.

The K-means of different metabolites (DAMs) were studied to investigate trends in the relative content of different metabolites in the three groups (Figure 8). The DAMs were classified into six clusters. The DAMs in Clusters 1, 3, and 6 were strongly detected in the RS group. Meanwhile, the concentration of DAMs was highest in the ZJ group in Cluster 5. Of note, a majority of DAMs in Cluster 6(32 out of 79 compounds) were flavonoids.

### 2.7. Pathway Enrichment Analysis of Differential Metabolites

We performed KEGG enrichment analysis on differential metabolites to identify the main metabolic pathways (Figure 9), and all metabolic pathways are shown in Appendix A. The obtained results revealed that the differential metabolites in the comparisons of these three groups were annotated in 20 pathways. Interestingly, the “flavonoid biosynthesis” and “flavone and flavonol biosynthesis” pathways were highly enriched in both the RS vs. ZJ and RS vs.GP comparison groups. In addition to the abovementioned two pathways, the GP vs. ZJ group was also enriched in the “glycerophospholipid metabolism” and “isoflavonoid biosynthesis.” In general, the “Flavone and flavonol biosynthesis” was the vital pathway generated by KEGG enrichment analysis.

### 2.8. Quantitative and Qualitative HPLC Analysis

The peak areas and standard concentrations of each flavonoid compound were linearly fitted to a linear relation of y=ax + b, where y represents the peak area and x represents the injection amount (μg) and measured by HPLC. As listed in Table 4, the linear relationships between the concentration and peak area of the four compounds were good (R^2^ ≥ 0.999). The precision RSDs of the four compounds were 0.97–2.94%. The values for repeatability were 2.01–2.46%. To confirm the stability, a standard solution mixed with methanol was analyzed at 0, 2, 4, 6, 8, 12, and 24 h to evaluate the stability of the solution, and the obtained results showed that the stability RSD ranged from 1.32 to 2.71%. The chromatogram of the four standards and the representative chromatogram of *D. officinale* used for qualification are shown in Figure 10. The obtained results indicated that the four flavonoids (Vicenin II, Vicenin I, Vicenin III and Rutin) fillings in the samples from the three production areas were different from one another (Figure 11). Among them, the highest total content of the four flavonoids was in the RS group with 415 μg/g; the content in GP and ZJ groups was lower, i.e., 323 μg/g and 259 μg/g (detailed data of four flavonoids are shown in Appendix A), respectively. Surprisingly, all four flavonoids showed a certain trend in their respective samples, i.e., Rutin had the highest content, followed by Vicenin II, Vicenin I, and Vicenin III.

## 3. Discussion

In our study, a widely targeted metabolomics approach was performed on wild-like *Dendrobium* stems of different origins. *D. officinale* metabolic analysis not only provided a vital basis for species identification but also provided a significant foundation for the quality control of wild-like *D. officinale*. The obtained results showed that 442 substances in 16 significant groups were detected, and the metabolism of *D. officinale* collected from different habitats greatly differed. Because several comparison groups were significantly enriched in the pathway of “secondary metabolite biosynthesis”, this study concentrated on the secondary metabolites of *D. officinale*.

In this study, the main metabolite species of wild-like *Dendrobium* were flavonoids, lipids, and amino acids and their derivatives. Cao et al. found that the concentrations of greenhouse *Dendrobium* metabolites, especially organic acids, amino acids and their derivatives, and nucleotides and their derivatives, were higher in the stems than in the leaves [23]. One limitation of previous research is that it typically focused on materials that were immature greenhouse plants of only 6 months old. Compared with our study, we used imitations of wild mature stems as our materials. Thus, it is reasonable to assume that most of the metabolites reported by the previous study were intermediate products. The significance of the previous report was in revealing that the small molecule metabolites of greenhouse *Dendrobium* stems and leaves were not significantly different, thus expanding the scope of application of *Dendrobium*. However, the use of these data as a comprehensive evaluation of medicinal *Dendrobium* was not rigorous. Interestingly, Yan Sui Mai et al. found that although greenhouse *D. officinale* protocorms (DOPs) and *D. officinale* caulis (Mat) had a similar chemical composition [26], they exhibited different patterns of content. Moreover, the differentially expressed genes between DOPs and Mat displayed that gene enrichment was mainly related to flavonoid biosynthesis. This study reinforced that there was a significant content difference between mature and immature stems of *Dendrobium*, and the content difference was mainly related to flavonoids, which is probably consistent with the results we showed. In summary, we selected mature plants grown in a simulated wild environment rather than those grown in greenhouses because mature samples were more compatible with practical utilization. In addition, by using wild-like *Dendrobium* as material, the medicinal value of *Dendrobium* can be better reflected.

The growing environment of plants affects the changes in metabolites in plants. Important factors affecting the accumulation of secondary metabolites in plants include different seasons, ages, and origins. Meng Q et al. investigated how different drying methods affect the characteristic components in *D. officinale* [27]. Zuo et al. conducted a comparative analysis of the metabolomics of *D. officinale* under different cultivation substrates [24]. They found that only three shared metabolites differed significantly among the different comparisons. Nevertheless, there were 22 significantly different metabolites shared among the three groups in our study. This difference occurred, and we speculated that the environmental factors differed greatly from one origin to another, thus leading to a greater variation in the metabolites of *D. officinale*. In contrast, the previous study, which only used three different cultivation substrates, had relatively little effect on *D. officinale* metabolites. A preliminary UPLC-Q-TOF-MS analysis comparing the compounds of *D. officinale* from different origins was performed by Juan Yang et al. [28]. They found that the chemical compositions of the samples in the same region are similar, while the chemical compositions of the samples in different regions are different. This finding was generally consistent with our research. Compared with our findings, we identified a greater number and a wider variety of metabolites than in previous studies. We revealed more in-depth differences in the metabolites of *Dendrobium* from different origins.

Flavonoids are important secondary metabolites, and flavonoids in medicinal plants have been shown to have a variety of biological functions [29,30,31,32,33]. The metabolomic profiling results suggested that flavonoid compounds accounted for the highest proportion of total metabolites, with 102 differential substances identified. Our cluster analysis of the three groups of differential metabolites (Figure 7a) revealed that flavonols were significantly upregulated in the RS group compared to the GP and ZJ groups. Flavonoids are widely found in the roots, stems, leaves, and flowers of *D. officinale* and have anti-cancer, antitumor [34], and antioxidant effects [35]. Therefore, we hypothesized that due to factors such as climate, environment or geographical location, *D. officinale* grown in Guangdong Province (RS) with a relatively high flavonol content had higher medicinal value. Meanwhile, flavonoids were the most variable metabolites in the three origins. Therefore, flavonoids were also used to evaluate *D. officinale* quality based on medicinal effectiveness in a previous study [36] because they perform many physiological functions.

The most abundant chemical constituent in *Dendrobium* is polysaccharide [37], which has multifaceted pharmacological properties such as antioxidant [38,39], antitumor [40], and hypolipidemic activities. However, in addition to polysaccharides, *D. officinale* also contains many secondary metabolites, such as flavonoids, lipids, and alkaloids, according to the results of metabolomic analysis, and these metabolites can often be the basis for distinguishing the quality of medicinal *D. officinale*. Therefore, it is particularly important to identify quality control markers beyond polysaccharides. To date, the *Chinese Pharmacopoeia* stock still regards the glucose-to-mannose peak area ratio as the quality control index of *D. officinale*. Our preliminary study found that some of the *Dendrobium huoshanense* glucose to mannose peak area ratios were also within this range [41], indicating that the *Chinese Pharmacopeia’s* method had some defects. Clearly, the *Chinese Pharmacopoeia* does not include differences in the production area of medicinal *D. officinale* in the quality indicators. W. Li et al. used metabolite profiling for Danshen in China to improve the quality control of Danshen-related preparations [42]. Similarly, our study used metabolomics techniques, and we selected four flavonoids that are stably present in *D. officinale*, established a simple and feasible method for their determination, and measured their contents. We were pleasantly surprised to find that the contents of the four compounds are different in *D. officinale* of the three origins. Among them, Vicenin II and Rutin [43,44] have certain pharmacological activities. Combined with our preliminary study [45,46], we hypothesized that they could be used as indicators for evaluating *D. officinale* quality.

The discovery of new makers was beneficial to improving the quality control of *D. officinale* in the *Chinese Pharmacopoeia*. With the help of rich knowledge of *Dendrobium* metabolites, a more cost-effective study can be conducted in the future for the substance accumulation and material accumulation of key metabolites of *Dendrobium* from different habitats. Multiomics technologies such as transcriptome, proteome, and genome have also been applied to the study of medicinal plants with the basis of genome sequences. Therefore, genomics, transcriptomics, and proteomics techniques can be combined to further study the secondary metabolites.

## 4. Materials and Methods

### 4.1. Plant Material

*Dendrobium officinale*, identified by Prof. Gang Wei (Guangzhou University of Chinese Medicine, Guangdong, China), was collected from Zhejiang Province (ZJ), Guangdong Province (RS), and Guangxi Province (GP) in China. All sample plants were grown in a local greenhouse under the conditions of a day temperature of 26 °C and a night temperature of 18 °C, with natural light. After 6 months, the plants were transferred to the wild to simulate their normal ecology. The experiment was conducted using at least 3-year-old wild-cultured *D. officinale*. We randomly selected three samples from each of the three groups for analysis, and 3 biological replicates of each origin were sampled in multiple mixed groups. As shown in Figure 12, ZJ, GP and RS samples all had rust-colored spots. Among them, the stems of the ZJ and GP samples were elongated, whereas the RS stems were short and thick. All stem samples were collected from three independent plants, immediately frozen in liquid nitrogen, and stored at −80 °C until used for metabolomic analysis.

### 4.2. Sample Preparation and Extraction

The freeze-dried leaves were crushed using a mixer mill (MM 400, Retsch) with zirconia beads for 1.5 min at 30 Hz. A total of 100 mg of powder was weighed and extracted overnight at 4 °C with 0.6 mL of 70% aqueous methanol. Following centrifugation at 10,000× *g* for 10 min, the extracts were absorbed (CNWBOND Carbon-GCB SPE Cartridge, 250 mg, 3 mL; ANPEL, Shanghai, China) and filtrated (SCAA-104, 0.22 μm pore size; ANPEL, Shanghai, China) before UPLC–MS/MS analysis.

### 4.3. UPLC-ESI-MS/MS

The prepared sample extraction was analyzed employing a UPLC–ESI–MS/MS system, including a UPLC system (Shim-pack UFLC SHIMADZU CBM30A, Shimadzu, Kyoto, Japan) and an ESI–MS/MS system (Applied Biosystems 4500 Q TRAP, AB SCIEX, Foster City, CA, USA). A separation operation was implemented using a 1.8 µm Agilent SB-C18 column (100 mm × 2.1 mm) equilibrated with mobile phases A and B prepared by mixing ultrapure water with 0.1% formic acid and acetonitrile, respectively. Chromatographic separation was completed by way of a gradient elution program. First, mobile phase B rose linearly from 5% to 95% within 9 min. It was maintained at the level of 95% from 9 to 10 min, decreased to 5% from 10 to 11 min, and held steady until 14 min. The temperature of the column oven was 40 °C, the injection volume was 4 µL, and the flow rate was 0.35 mL/min [47]. The effluent was alternatively connected to an ESI–triple quadrupole-linear ion trap (QTRAP)-MS.

### 4.4. ESI-Q TRAP-MS/MS Analysis

Linear ion trap (LIT) and triple quadrupole (QQQ) scans were acquired on a triple quadrupole-linear ion trap mass spectrometer (Q TRAP), API 4500 Q TRAP UPLC/MS/MS System, equipped with an ESI Turbo Ion-Spray interface, operating in positive and negative ion mode and controlled by Analyst 1.6.3 software (AB Sciex). The main parameters were as follows: the temperature of electrospray ionization was set to 550 °C; the voltages of the ion sprays were 5500 V and −4500 V in positive and negative ion modes, respectively; the curtain gas (CUR) was 30 psi, and the collision-activated dissociation (CAD) was high. Each ion pair was scanned under the conditions of the optimum declustering potential (DP) and collision energy (CE) in the QQQ system [48]. Instrument tuning and mass calibration were performed with 10 and 100 μ mol/L polypropylene glycol solutions in QQQ and LIT modes, respectively. QQQ scans were acquired as MRM experiments with collision gas (nitrogen) set to 5 psi. DP and CE for individual MRM transitions were performed with further DP and CE optimization. A specific set of MRM transitions was monitored for each period according to the metabolites eluted within this period.

### 4.5. Metabolic Profiling of D. officinale Stems

According to the metabolite information public database and self-built catalog MWDB (Met Ware Biotechnology Co., Ltd. Wuhan, China), the substance is qualitatively based on the secondary spectrum data, and the isotope signal is detached during the analysis, including the repetitive signals of K^+^, NH_4_^+^, Na^+^, and repetitive signals of fragment ions of other greater molecular weight substances. Then a triple quadrupole is used to screen out the characteristic ions of each substance, and the indicator intensity of the representative ions is gained in the indicator. MultiaQuant software 3.0.3 was used to open the MS file under the sample to integrate and calibrate the chromatographic peaks. The relative content of the corresponding substance in the peak part of the apparent chromatographic peak can be calculated. To conclude, the integral data of all chromatographical peak areas can be obtained. To measure the content of each metabolite in samples from different origins, we standardized the MS peaks found by separate metabolites in samples from different origins based on the preservation time and peak decoration of the metabolites. Therefore, the accuracy of quantitative and qualitative analysis is additionally guaranteed.

### 4.6. Data Analysis

To simplify the complicated data gained from the abovementioned methods, we utilized principal component analysis (PCA)and orthogonal partial least squares discriminant analysis (OPLS-DA). Data analysis was conducted mainly using R software. PCA was performed by the built-in “prcomp” function (Version 3.5.0, University of Auckland, Auckland, New Zealand), and the data were processed by unit variance scaling before unsupervised PCA. The hierarchical cluster analysis (HCA) consequences of samples and metabolites are displayed in the form of a dendrogram heatmap. For HCA, the normalized signal intensity of the metabolites is visualized as a chromatogram. OPLS-DA was conducted on the raw data by the statistics function “MetaboAnalystR” [48] (Version 1.0.1, McGill University, Quebec, Canada). The metabolites that were significantly regulated among groups were determined by variable importance in projection (VIP ≥ 1) and fold-change (|log2 (FC)| ≥ 1). They were transformed with the log2 function and mean-centered. To avoid overfitting, a permutation test was performed. The identified metabolites were annotated based on the KEGG compound database, and the annotated metabolites were then matched to the KEGG pathway database. Pathways that significantly regulated metabolites mapped to were fed into a metabolite set enrichment analysis (MSEA), and their significance was calculated by the *p*-values from hypergeometric trials.

### 4.7. HPLC Quantitative and Qualitative Analysis Conditions

Quantitative analysis was performed using an Agilent 1200 system (Agilent, Santa Clara, CA, USA). Chromatographic separations were performed on a Kromasil 100–5 C18 column (250 × 4.6 mm, 5 μm; Agilent, Santa Clara, CA, USA) maintained at 40 °C. The mobile phase consisted of acetonitrile and methanol (mobile phase A, *v*/*v* = 1:1) and 10 mM ammonium acetate buffer (mobile phase B) (with a gradient elution program of 0–20 min, A: 14–18%, B: 86–82%; 20–35 min, A: 18–22%, B: 82–78%; 35–45 min, A: 22%~26%, B: 78–74%; 45–55 min, A: 26–30%, B: 74–70%), at a flow rate of 0.8 mL/min. The injection volume was 10 μL each time. The detection wavelength was set to 340 nm. Four standard reference materials were used to help identify chromatographic peaks. Among them, Vicenin I was previously isolated by our research team (Gang Wei, Guangzhou University of Chinese Medicine), and the other three materials (Rutin, Vicenin II, and Vicenin III) were purchased from Shanghai Standard Technology Co., Ltd. (Shanghai, China).

## 5. Conclusions

In this experiment, the metabolome profiles of *D. officinale* in three main origins (RS, GP and ZJ) were researched based on the UPLC-ESIMS/MS method, and a total of 442 different metabolites were identified. Compared with these three origins, more flavonoids accumulated in the RS samples; furthermore, the plant appearance of the RS samples was most similar to the wild. We speculated that the relatively optimal source of origin of wild-like *D. officinale* was from the Guangdong Province. Although the samples from all three sources contained similar metabolites, their contents greatly differed, with most of them belonging to flavonoids. In addition, “flavonoid biosynthesis” and “flavone and flavonol biosynthesis” were significantly affected by the fluctuating metabolites in the KEGG pathway enrichment analysis, as well as the differences in the biosynthetic pathways mainly based on apigenin and quercetin as parent nuclei. It is proved that the metabolites of *D. officinale* in a different origin carry the information of their origin, and the difference of metabolites is feasible for distinguishing *D. officinale* origins. Meanwhile, the HPLC analysis of the four flavonoids in Dendrobium revealed that Rutin had the highest content. Consequently, we speculated that Rutin and Vicenin II could be used as new markers for the quality evaluation of *D. officinale*. In summary, there is an origin effect on the accumulation of metabolites in *D. officinale,* and we provided important insights into searching for signature compounds for the quality evaluation of *D. officinale.*

## Figures and Tables

**Figure 1 molecules-27-07024-f001:**
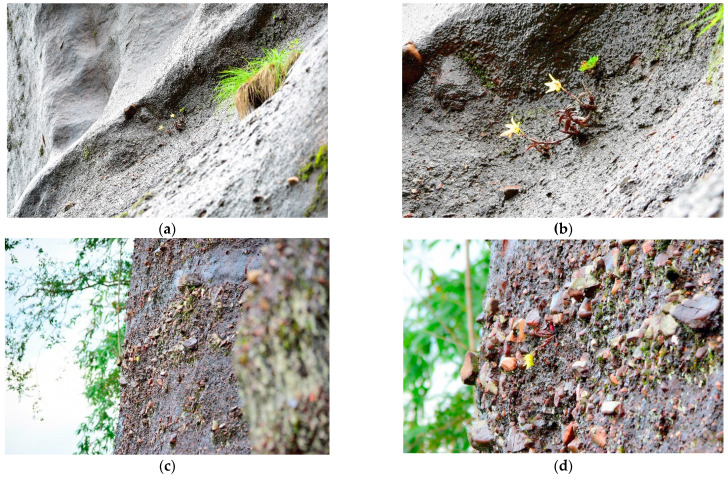
The ecological environment of wild *D. officinale*. Some of the wild *D. officinale* grows on the cliffs at an altitude of 274.5 m, and the geographic location is 25°1′27″ N, 113°1′44″2 E. (**a**) Two clusters of wild *D. officinale* growing on a stone wall in the distance. (**b**) Close up view of two clusters of wild *D. officinale* growing on the stone wall. (**c**) One cluster of wild *D. officinale* growing on a stone wall in the distance. (**d**) Close up view of a cluster of wild *D. officinale* growing on a stone wall. The researchers did not cause damage to wild *D. officinale* by long-distance shooting.

**Figure 2 molecules-27-07024-f002:**
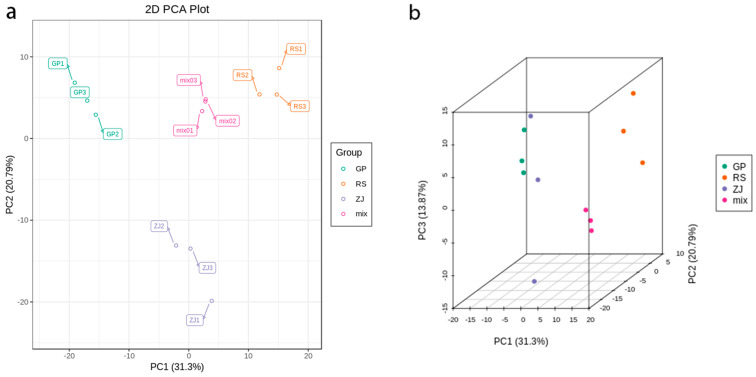
Score plot of PCA for *D. officinale* samples: (**a**) 2D score plot of PCA for *D. officinale* samples originate from GP, ZJ, RS, and MIX samples. PC1 represents the first principal component, and PC2 represents the second principal component. (**b**) 3D score plot of PCA for *D. officinale* samples originate from GP, ZJ, RS, and MIX samples. PC3 represents the third principal component.

**Figure 3 molecules-27-07024-f003:**
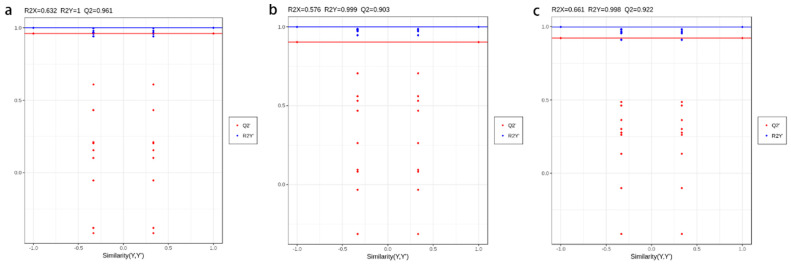
The model verification of OPLS-DA: (**a**) RS vs. GP; (**b**) RS vs. ZJ; (**c**) GP vs. ZJ. Horizontal lines correspond to the original model of R2Y and Q2, and the red and blue dots represent Q2’ and R2Y’ of the model after replacing Y, respectively. R2Y is the interpretation rate of the model to the Y matrix (Class label), and Q2 is the predictive ability of the model.

**Figure 4 molecules-27-07024-f004:**
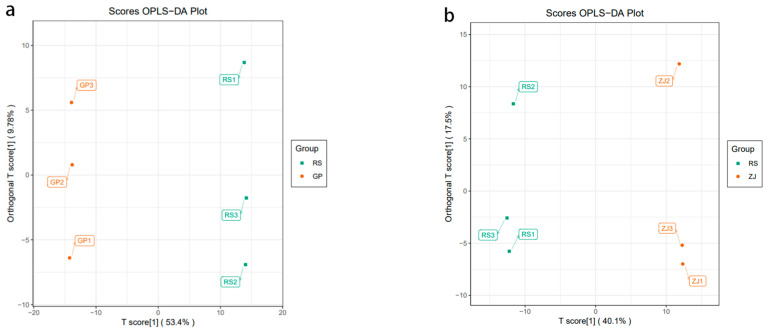
Score plots of OPLS-DA: (**a**) RS vs. GP; (**b**) RS vs. ZJ; (**c**) ZJ vs. GP. The model is valid when Q^2^ > 0.5. The *X*-axis is the principal predictive component, and the *Y*-axis is the principal orthogonal component.

**Figure 5 molecules-27-07024-f005:**
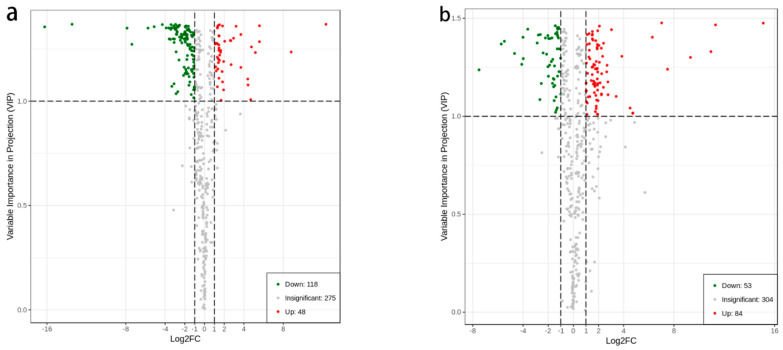
Volcano plots: (**a**) RS vs. GP; (**b**) GP vs. ZJ; (**c**) RS vs. ZJ. The green dots in the plots illustrated that the differential metabolites were significant and down-regulated, while the red dots illustrated that the differential metabolites were significant but upregulated, and the black dots illustrated that the metabolites could be detected but not significantly different.

**Figure 6 molecules-27-07024-f006:**
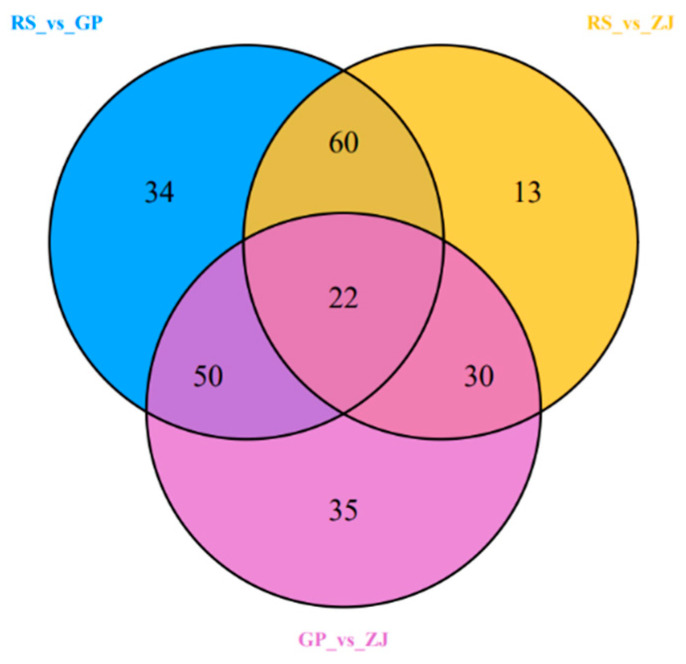
The Venn diagram illustrates shared or unique metabolites that differed significantly in terms of content among the different comparisons.

**Figure 7 molecules-27-07024-f007:**
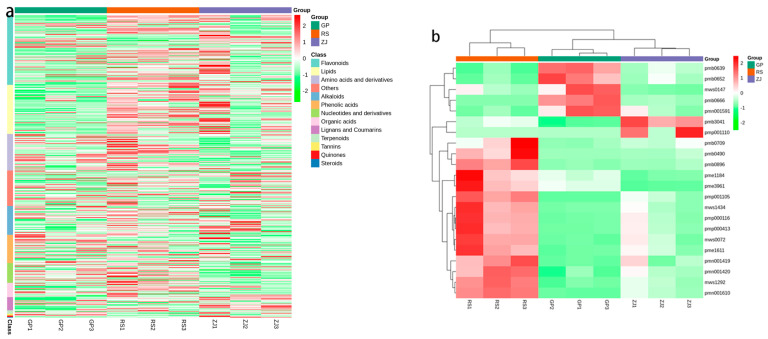
Heatmaps of hierarchical cluster analysis (HCA): (**a**) GP vs. RS vs. ZJ; (**b**) The 22 different metabolites commonly found in RS, GP and ZJ samples. The abscissa is used to display the names of samples, and the ordinate on the right is used to display the classes or names of metabolites. The deeper the red color, the higher the content of the metabolites; the deeper the green color, the lower the content of the metabolites.

**Figure 8 molecules-27-07024-f008:**
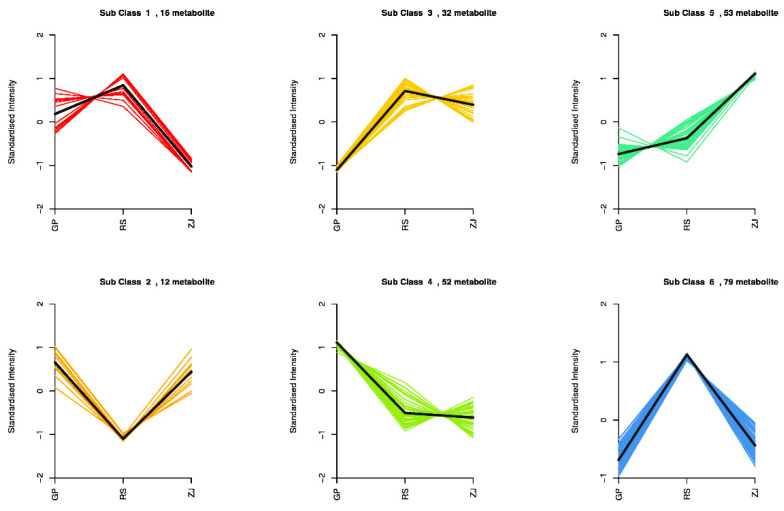
The K means analysis of different metabolites. The horizontal coordinate represents the sample name, and the vertical coordinate represents the standardized metabolite relative content. The Sub Class represents the number of metabolite classes with the same trend of change.

**Figure 9 molecules-27-07024-f009:**
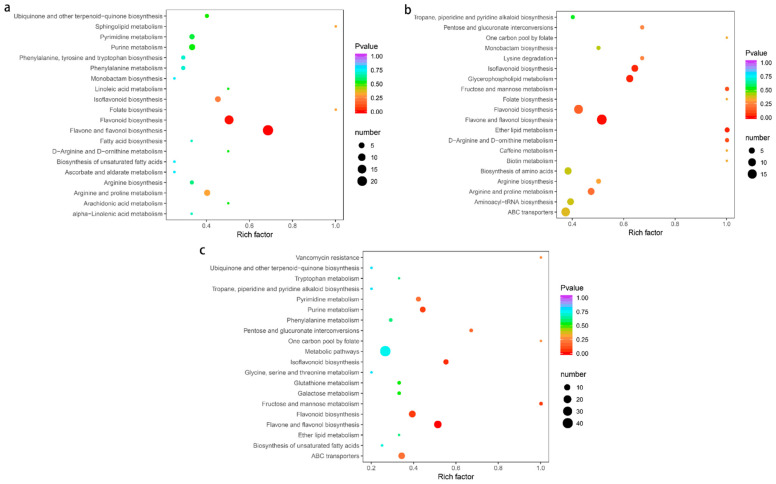
KEGG enrichment maps of differential metabolites: (**a**) RS vs. GP; (**b**) GP vs. ZJ; (**c**) ZJ vs. RS. The abscissa represents the enrichment factor of the pathway, and the ordinate shows the names of pathways. The color of the dot represents the *p*-value, and the deeper the red of the dot, the stronger the enrichment effects. The size of points represents the number of metabolites enriched in the pathways.

**Figure 10 molecules-27-07024-f010:**
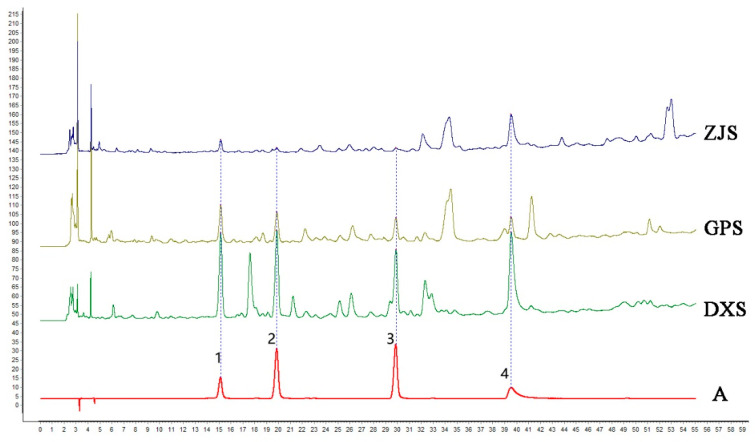
Qualitative analysis in the extracts of *D. officinale*: A represents the HPLC chromatogram of the four standards (1. Vicenin II; 2. Vicenin I; 3. Vicenin III; 4. Rutin.). DXS, GPS and ZJS represent RS samples, GP samples and ZJ samples, respectively.

**Figure 11 molecules-27-07024-f011:**
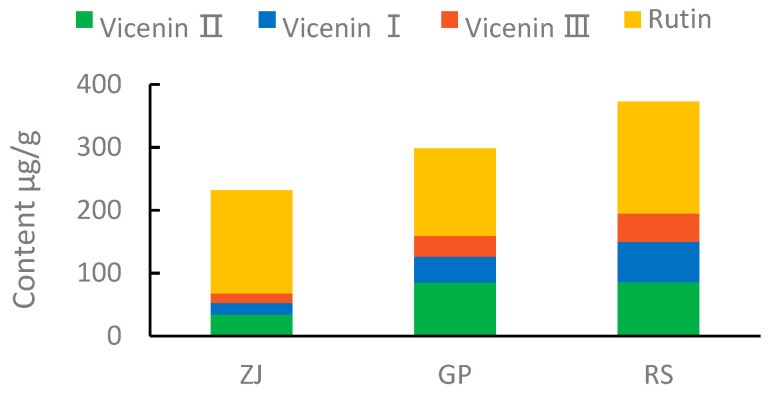
Content of four main flavones in ZJ, GP, and RS samples, including Vicenin II, Vicenin I, Vicenin III, and Rutin.

**Figure 12 molecules-27-07024-f012:**
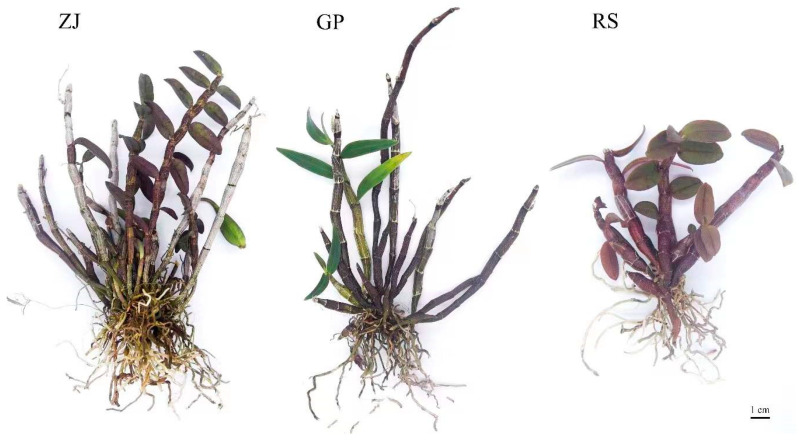
The appearance of *D. officinale* from ZJ, GP, and RS, respectively. GP represents Guiping, Guangxi Province, RS represents Shaoguan, GP represents Guangdong Province, and ZJ represents Zhejiang Province.

**Table 1 molecules-27-07024-t001:** Average Pearson’s correlation coefficients for comparisons of intragroup and intergroup samples.

Group	Average Pearson’s Correlation Coefficients *
GP vs. GP	0.94421
RS vs. RS	0.97132
ZJ vs. ZJ	0.95574
GP vs. RS	0.83320
RS vs. ZJ	0.92307
ZJ vs. GP	0.82796

The reproducibility of samples and the stability of the instruments are defined as good when the correlation coefficients of intragroup comparisons are higher than intergroup comparisons. The data were considered reliable when the correlation coefficients of intragroup samples were over 0.8.

**Table 2 molecules-27-07024-t002:** Classification of the 442 detected metabolites in *D. officinale* stems from three habitats.

Class	Number of Compounds	Class	Number of Compounds
Flavonoids	102	Others	20
Lipids	72	Lignans and Coumarins	19
Amino acids and their derivatives	53	Vitamins	7
Alkaloids	42	Terpenoids	5
Phenolic acids	41	Proanthocyanidins	2
Nucleotides and derivatives	29	Quinones	2
Carbohydrates and Alcohols	25	Tannins	1
Organic acid	21	Steroids	1

**Table 3 molecules-27-07024-t003:** A total of 22 different metabolites shared metabolites that differed significantly in terms of content among the RS vs. GP vs. ZJ group.

Compounds	Class
Kaempferol-3-neohesperidoside-7-glucoside	Flavonoids
Apigenin-6-C-β-D-xyloside-8-C-β-Darabinoside	Flavonoids
Apigenin 5-O-glucoside	Flavonoids
Isoschaftoside	Flavonoids
Isovitexin	Flavonoids
8-C-Hexosyl-apigenin O-hexosyl-O-hexoside	Flavonoids
C-Hexosyl-apigenin O-pentoside	Flavonoids
6-C-Hexosyl-apigenin O-sinapoylhexoside	Flavonoids
Quercetin 7-O-malonylhexosyl-hexoside	Flavonoids
Tricin O-saccharic acid	Flavonoids
Isohemiphloin	Flavonoids
Apigenin-8-C-glucoside	Flavonoids
Genistein 8-C-glucoside	Flavonoids
2,3-Dihydroxybenzoic Acid	Organic acids
3-Hydroxy-3-methyl butyric acid	Organic acids
N’-Feruloyl putrescine	Alkaloids
N-*p*-Coumaroyl putrescine	Alkaloids
Deoxyguanosine	Nucleotides and derivatives
Deoxyadenosine	Nucleotides and derivatives
1-O-[(E)-*p*-Cumaroyl]-β-D-glucopyranose	Phenolic acids
1-O-[(E)-Caffeoyl]-β-D-glucopyranose	Phenolic acids
Rutundic acid	Terpenoids

**Table 4 molecules-27-07024-t004:** Investigation results of the linear relationship of the HPLC methods for four flavonoids.

Components	Linear Equations	R^2^	Repeatability RSD (%)	Precision RSD (%)	Stability RSD (%) (n = 7)
Rutin	y = 14.997x−127.4	0.9997	2.46	2.69	2.41
Vicenin I	y = 16.656x−39.748	0.9999	2.01	2.94	2.71
Vicenin II	y = 12.589x−27.568	0.9998	2.04	0.97	1.32
Vicenin III	y = 18.269x−35.367	0.9999	2.45	2.03	2.52

## Data Availability

Not applicable.

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
