# Peer review of "Does the Metabolome of Wild-like Dendrobium officinale of Different Origins Have Regional Differences?"

_molecules, 2022, doi:10.3390/molecules27207024_

Round 1
Reviewer 1 Report
1. In l.3, what does the term "metabolite" mean? Should it be "metabolome"?
2. In l.156, the phrase "during the different production areas" is unclear. Please explain.
3. It is emphasized throughout the text that flavonoids played a central role in the discrimination process. The authors should state whether a statistical analysis based solely on flavonoids (and other polyphenols) would be a credible tool for such a study.
4. How was Cis-4,7,10,13,16,19-docosahexaenoic acid identified (l.183).
5. In the discussion section, l.285-296 are rather redundant and could be added in Introduction.
6. In the conclusions, it is stated that rutin and vicenin could be used as markers of differentiation. Is this supported by the study? How is it substantiated?
Author Response
Response
Dear Reviewer #1,
Our manuscript titled was “Does the metabolome of Wild-like Dendrobium officinale of different origins have regional differences?” and the manuscript ID: molecules-1961071.We would like to thank you for your careful reading, helpful comments, and constructive suggestions, which have significantly improved the presentation of our manuscript.The responses to the reviewer’s comments were as follows:
- In l.3, what does the term "metabolite" mean? Should it be "metabolome"?
The authors’ answer: We gratefully appreciate your valuable suggestion. We have rewritten this part according to your suggestion. We have changed the term "metabolite" to "metabolome".
- In l.156, the phrase "during the different production areas" is unclear. Please explain.
The authors’ answer: Thank you for pointing out this problem in the manuscript. We didn't articulate the point well, the phrase "during the different production areas" meant D. officinale from three different origins, including Zhejiang Province (ZJ), Guangdong Province (RS) , and Guangxi Province (GP).
l.164 (l.156 in the original manuscript), modified: "To study the variations in metabolites from the different origins of D. officinale, "
- It is emphasized throughout the text that flavonoids played a central role in the discrimination process. The authors should state whether a statistical analysis based solely on flavonoids (and other polyphenols) would be a credible tool for such a study.
The authors’ answer: Thank you for the above suggestion. We referred to a formulation from the paper [1] that indicated that "The metabolomic profiling results suggested that flavonoid compounds had maximum counts of metabolites in D. officinale. Flavonoids were also the metabolites with the greatest variation in the three cultivation substrates. For this reason, the flavonoids could be used to evaluate the quality of D. officinale.". Their research concluded that flavonoids could be used to evaluate the quality of D. officinale, and play an important role in the discrimination process of different cultivation substrates.
According to our previous research [2], "HPLC coupled with diode-array detection and HPLC multiple-stage tandem mass spectrometry was used to identify the chemical constituents of D. nobile from various habitats", and we found that "The D. nobile habitats were distinguished by significant differences in their flavone content. The C-glycosyl flavones were demonstrated to be characteristic compounds for evaluating D. nobile from various habitats. ". Furthermore, in our study, we performed principal component analysis (PCA) on the overall metabolome of D. officinale from the three different origins. The results indicated that the three groups of samples were well distinguished (I.136-137). Therefore, combined with previous studies, we would like to express that the technique of using widely targeted metabolomics can be a reliable tool to distinguish the origins. Flavonoids play an important role in differential metabolites.
Reference:
[1] Zuo, S.; Yu, H.; Zhang, W.; Zhong, Q.; Chen, W.; Chen, W.; Yun, Y.; Chen, H. Comparative Metabolomic Analysis of Dendrobium officinale under Different Cultivation Substrates. Metabolites. 2020, 10, 325. https://doi.org/10.3390/metabo10080325.
[2] Wang, Y.W.; Liao, X.; Zhou, C.J.; Hu, L.; Wei, G.; Huang, Y.C.; Lei, Z.X.; Ren, Z.R.; Liu, Z.X.; Liu, Z.H. Identification of C-glycosyl flavones and quality assessment in Dendrobium nobile. Rapid Commun Mass Spectrom. 2021, 35, e9012. https://doi.org/10.1002/rcm.9012.
- How was Cis-4,7,10,13,16,19-docosahexaenoic acid identified (l.183).
The authors’ answer: Thank you for pointing out this problem in the manuscript. According to the check of the assay analyst, cis-4,7,10,13,16,19-docosahexaenoic acid was identified by UPLC–ESI–MS/MS method. According to the metabolite information public database and self-built catalog MWDB (Met Ware Biotechnology Co., Ltd. Wuhan, China), the substance is qualitatively based on the secondary spectrum data, and the isotope signal is detached during the analysis, including the repetitive signals of K+, NH4+, Na+, and repetitive signals of fragment ions of other greater molecular weight substances. The secondary spectrum data and retention times of the substance were verified to be consistent with the database information, the result was shown in the following figure.(Please see the attachment, the figure can not be inserted here,)
- In the discussion section, l.285-296 are rather redundant and could be added in Introduction.
The authors’ answer: Thank you for the above suggestions. According to your suggestion, we have changed the formulation of this paragraph to be more concise and added it to the Introduction.
l.62-69 (l.285-294 in original manuscript), added: Metabolomic approaches provide valuable resources for the identification of metabolite differences between mutant and wild-type plants, the identification of metabolite changes in original herbs, and the quality control of herbal medicines [13-17]. Metabolomics is based on high-throughput tools that allow the simultaneous qualitative and quantitative analysis of many metabolites and the discovery of correspondences between metabolites and physiological changes [18]. Therefore, it can be used to detect changes in metabolites in plants under different growth environments.
l.310-312 (l.294-296 in original manuscript), modified: D. officinale metabolic analysis not only provided a vital basis for species identification but also provided a significant foundation for the quality control of wild-like D. officinale.
- In the conclusions, it is stated that rutin and vicenin could be used as markers of differentiation. Is this supported by the study? How is it substantiated?
The authors’ answer: We totally understand the reviewer’s concern. We feel sorry for the inconvenience brought to the reviewer.
(1) In l.516-517, "Consequently, we speculated that Rutin and Vicenin â…¡ can be used as new markers for the quality control of D. officinale.", we wanted to emphasize markers of quality evaluation rather than markers of differentiation. In l.366-368, "Therefore, flavonoids were also used to evaluate D. officinale quality based on medicinal effectiveness in a previous study [36], because they perform many physiological functions. ", Yang et al. believed that flavonoids could be used to evaluate D. officinale quality. Kroslakova et al. indicated that flavonoids were a widespread group of phytochemistry with diverse biological functions and significant substances in plants, which have served as excellent chemical markers to control the quality of medicinal plants [1]. In our study, both Rutin and Vicenin â…¡ belonged to flavonoids.
(2) In addition to this manuscript, we had other researches analyzing the Rutin and Vicenin â…¡ in D. officinale. In our previous study [2], a method was established to determine the content of Vicenin â…¡, and we analyzed the variations of Vicenin â…¡ content among 26 batches from different habitats. Furthermore, we also conducted analysis on 32 batches of D. officinale samples of different origins, and we found that Vicenin â…¡ was a relatively stable common peak in different source samples, and the characteristic peaks of Rutin was quite different [3]. Our previous study suggested that Vicenin â…¡ was suitable to be a reference peak for characteristic chromatogram, and the relative abundance of rutin peaks could be used as a reference to judge the categories of D. officinale.
(3) At last, combined with our preliminary studies, we hypothesized that Rutin and Vicenin â…¡ can be used as markers for evaluating D. officinale quality.
l.386-388, we have added two references ([45,46]) to the manuscript, and modified: Among them, Vicenin â…¡ and Rrutin [43,44] have certain pharmacological activities, combined with our preliminary study [45,46], we hypothesized that which indicates that they can be used as indicators for evaluating D. officinale quality.
Reference:
[1] Kroslakova, I.; Pedrussio, S.; Wolfram, E. Direct Coupling of HPTLC with MALDI-TOF MS for Qualitative Detection of Flavonoids on Phytochemical Fingerprints. Phytochemical Analysis. 2016, 27, 222–228. https://doi.org/10.1002/pca.2621
[2] Wang, Y.W.; Liao, X.; Zhou, C.J.; Hu, L.; Wei, G.; Huang, Y.C.; Lei, Z.X.; Ren, Z.R.; Liu, Z.X.; Liu, Z.H. Identification of C-glycosyl flavones and quality assessment in Dendrobium nobile. Rapid Commun Mass Spectrom. 2021, 35, e9012. https://doi.org/10.1002/rcm.9012.
[3] Liang, Z.Y.; Xie, Z.S.; Huang, Y.C.; Yuan, Y.; Zhou, C.J.; Wang, Y.W.; Wei, G. HPLC Characteristic Spectrum Optimization of Flavonoid Glycosides on Dendrobium officinale and Characteristics Analysis of Different Provenances. Chinese Journal of Experimental Traditional Medical Formulae. 2019, 25, 22-28. https://doi.org/10.13422 /j.cnki.syfjx.20190112.
We tried our best to improve the manuscript and made some changes. We appreciate for Reviewer’s warmwork earnestly, and hope that the correction will meet with approval. Once again thank you very much for your comments and suggestions.
Sincerely,
Gang Wei, weigang021@outlook.com
Huan Wang, wanghuanchanyeol@163.com
Corresponding authors.

Reviewer 2 Report
The manuscript entitled ‘Does the metabolite of Wild-like D. officinale of different origins have regional differences?’ addresses the application of the widely targeted metabolomics approach for the assessment of the metabolite profile of Dendrobium officinale obtained from plant stems of different origin. This manuscript provides a contribution in the quality study of herbal medicines.
The manuscript is composed according the scientific methodology and data are presented appropriately. In the introductory part authors provided the basic information related to the study and cited sufficient number of relevant references.
Results of the study are presented as tables, graphs and figures of sufficient resolution, and in certain extent support the conclusions.
Although the manuscript is scientifically sound and methodologically correct, in my opinion it has some inaccuracies especially in the introductory part and the results and discussion section, therefore it must be corrected. English language corrections are needed as well.
Below some points are indicated:
In the manuscript title: D.officinale should be replaced with Dendrobium officinale.
In the abstract section:
The sentence below is unclear:
the metabolite of Dendrobium officinale is an important factor for its pharmacological activity, but its metabolite is greatly influenced by environmental factors in 14 different habitats.
In the Introduction section:
The sentence is unclear: ...in anti-diabetes, anticancer, antioxidant, and anti-fatigue grown in different origins and..
The sentence should be reformulated: It is vital to comprehensively evaluate D. officinale.
The authors should explain which type of polysaccharides are the most abundant. In the manuscript they have indicated that the most abundant chemical constituent in Dendrobium is polysaccharide’ and they are the main sources of benefits generated by D. officinale, with pharmacological properties such as antioxidant, antitumor, and hypolipidemic activities. However polycaccharides are not mentioned within the 16 classes of metabolites!
Author Response
Response
Dear Reviewer #2,
Our manuscript titled was “Does the metabolome of Wild-like Dendrobium officinale of different origins have regional differences?” and the manuscript ID: molecules-1961071.We would like to thank you for your careful reading, helpful comments, and constructive suggestions, which have significantly improved the presentation of our manuscript. The responses to the reviewer’s comments were as follows:
Reviewer #2:
The manuscript entitled ‘Does the metabolite of Wild-like D. officinale of different origins have regional differences?’ addresses the application of the widely targeted metabolomics approach for the assessment of the metabolite profile of Dendrobium officinale obtained from plant stems of different origin. This manuscript provides a contribution in the quality study of herbal medicines.
The manuscript is composed according the scientific methodology and data are presented appropriately. In the introductory part authors provided the basic information related to the study and cited sufficient number of relevant references.
Results of the study are presented as tables, graphs and figures of sufficient resolution, and in certain extent support the conclusions.
Although the manuscript is scientifically sound and methodologically correct, in my opinion it has some inaccuracies especially in the introductory part and the results and discussion section, therefore it must be corrected. English language corrections are needed as well.
The authors’ answer: We thank the reviewer for reading our paper carefully and giving the above positive comments. We are sorry for the spelling mistakes and grammatical errors caused by our carelessness. In the revised version, we have made significant efforts to remove the mistakes and errors. All the errors you picked and the recommendations you proposed are greatly helpful for us to polish our manuscript. We appreciate your elaborate efforts in reviewing. Below we have attached the language editing certificate.(Please see the attachment, the figure could not be inserted here.)
Below some points are indicated:
- In the manuscript title: officinale should be replaced with Dendrobium officinale.
The authors’ answer: Thank you so much for your careful check. We have made correction according to your comments. (Lines 2 in the manuscript)
- In the abstract section:
The sentence below is unclear:
the metabolite of Dendrobium officinale is an important factor for its pharmacological activity, but its metabolite is greatly influenced by environmental factors in different habitats.
The authors’ answer: Thank you for pointing out this problem in the manuscript. We have rewritten this part.
l.14-16, modified: Environmental factors of different origins have a great influence on Dendrobium officinale metabolites, which affect its pharmacological activity.
- In the Introduction section:
The sentence is unclear: ...in anti-diabetes, anticancer, antioxidant, and anti-fatigue grown in different origins and.
The authors’ answer: Thank you so much for your careful check. We have rewritten the sentence.
l.32-34, modified: D. officinale is rich in polysaccharides, flavonoids, alkaloids and other bioactive substances, which play important pharmacological roles in anti-diabetic, anti-cancer, antioxidant and anti-fatigue.
l.55-56, modified: Therefore, it is important to explore the differences in the composition of D. officinale from different origins and identify the optimal growth environment to improve yield and quality.
The sentence should be reformulated: It is vital to comprehensively evaluate D. officinale.
The authors’ answer: Thank you for your comments. We have reformulated the sentence.
l.89-91, modified: By comparing their chemical compositions from the three main origins, the differences in their compositions were revealed, which was vital to comprehensively evaluate D. officinale.
- The authors should explain which type of polysaccharides are the most abundant.
In the manuscript they have indicated that the most abundant chemical constituent in Dendrobium is polysaccharide’ and they are the main sources of benefits generated by D.officinale, with pharmacological properties such as antioxidant, antitumor, and hypolipidemic activities. However, polycaccharides are not mentioned within the 16 classes of metabolites!
The authors’ answer: Thank you for the above suggestion. We totally understand the reviewer’s concern. Our group has conducted long-term research on D. officinal stem and found that crude polysaccharide was the most abundant [1]. The Chinese Pharmacopoeia stock specified that the content of polysaccharides should not be less than 25%. The polysaccharides of D. officinal were mainly composed of glucose and mannose, and sometimes contained less arabinose and rhamnose [2,3], with molecular weights ranging from tens of thousands to millions of Da [4]. The possible structure of the repeating unit of two polysaccharide fractions could be suggested as shown in the figure. In our manuscript, the widely targeted metabolic profiling detected small molecular weight substances, typically no larger than 1.5 kDa. Combining the articles above, we suggested that the molecular weights of polysaccharides are not in the scope of the detection. Therefore, polysaccharides were not mentioned within the 16 classes of metabolites.
Note: This figure was taken from our previous study [1].(Please see the attachment, the figure could not be inserted here.)
Reference:
[1] Tao, S.C.; Lei, Z.X.; Huang, K.W.; Li, Y.R.; Ren, Z.Y.; Zhang, X.F.; Wei, G.; Chen, H.M. Structural characterization and immunomodulatory activity of t-wo novel polysaccharides derived from the stem of Dendrobium officinale Kimura et Migo. Journal of Functional Foods. 2019, 57, 121-134. https://doi.org/10.1016/j.jff.2019.04.013.
[2] He, T.B.; Huang, Y.P.; Yang, L.; Liu, T.T.; Gong, W.Y.; Wang, X.J.; Sheng, J.; Hu, J.M. Structural characterization and immunomodulating activityof polysaccharide from Dendrobium officinale. International Journal of Biological Macromolecules. 2016, 83, 34-41. http://dx.doi.org/10.1016/j.ijbiomac.2015.11.038
[3] Lin, X.; Shaw, P.C.; Sze, S.C.; Tong, Y.; Zhang, Y.B.; Dendrobium oficinale polysaccharides ameliorate the abnormality ofaquaporin 5, pro-inflammatory cytokines and inhibit apoptosis in the experimental Sjogren's syndrome mice.Int Immunopharmacol. 2011, 11, 2025-2032. http://doi:10.1016/j.intimp.2011.08.014.
[4] Zhang, X.F; Duan, S.N.; Tao, S.C.; Huang, J.H.; Liu, C.X.; Xing, S.P.; Ren, Z.Y.; Lei, Z.X.; Li, Y.R.; Wei, G. Polysaccharides from Dendrobium officinale inhibit proliferation of osteosarcoma cells and enhance cisplatin-induced apoptosis. Journal of Functional Foods. 2020, 73, 104143. https://doi.org/10.1016/j.jff.2020.104143
We tried our best to improve the manuscript and made some changes. We appreciated for Reviewer’s warm work earnestly and hoped that the correction will meet with approval. Once again thank you very much for your comments and suggestions.
Sincerely,
Gang Wei, weigang021@outlook.com
Huan Wang, wanghuanchanyeol@163.com
Corresponding authors.

Round 2
Reviewer 2 Report
Authors addressed the main questions and in my opinion after minor corrections the manuscript can be accepted for the publication.